# EpiExploreR: A Shiny Web Application for the Analysis of Animal Disease Data

**DOI:** 10.3390/microorganisms7120680

**Published:** 2019-12-11

**Authors:** Lara Savini, Luca Candeloro, Samuel Perticara, Annamaria Conte

**Affiliations:** Centro Operativo Veterinario per l’Epidemiologia, Programmazione, Informazione e Analisi del Rischio (COVEPI), National Reference Center for Veterinary Epidemiology, Istituto Zooprofilattico Sperimentale, dell’Abruzzo e del Molise “G. Caporale”, Campo Boario, 64100 Teramo, Italy; l.candeloro@izs.it (L.C.); s.perticara@izs.it (S.P.); a.conte@izs.it (A.C.)

**Keywords:** R-software, Shiny, spatiotemporal analyses, zoonosis, vector borne diseases, SaTScan, network analysis

## Abstract

Emerging and re-emerging infectious diseases are a significant public and animal health threat. In some zoonosis, the early detection of virus spread in animals is a crucial early warning for humans. The analyses of animal surveillance data are therefore of paramount importance for public health authorities to identify the appropriate control measure and intervention strategies in case of epidemics. The interaction among host, vectors, pathogen and environment require the analysis of more complex and diverse data coming from different sources. There is a wide range of spatiotemporal methods that can be applied as a surveillance tool for cluster detection, identification of risk areas and risk factors and disease transmission pattern evaluation. However, despite the growing effort, most of the recent integrated applications still lack of managing simultaneously different datasets and at the same time making available an analytical tool for a complete epidemiological assessment. In this paper, we present EpiExploreR, a user-friendly, flexible, R-Shiny web application. EpiExploreR provides tools integrating common approaches to analyze spatiotemporal data on animal diseases in Italy, including notified outbreaks, surveillance of vectors, animal movements data and remotely sensed data. Data exploration and analysis results are displayed through an interactive map, tables and graphs. EpiExploreR is addressed to scientists and researchers, including public and animal health professionals wishing to test hypotheses and explore data on surveillance activities.

## 1. Introduction

Emerging and re-emerging infectious diseases are a significant public and animal health threat, and their early detection and immediate response are crucial for their control. The detection of an outbreak or an increase of cases of a zoonotic disease (e.g., West Nile Fever, Brucellosis) in animals could be the first signal for public health authorities to start the implementation of prevention programs [1,2,3]. The analyses of animal surveillance data might therefore be of paramount importance for public health authorities to identify the appropriate control measure and intervention strategies in case of epidemics. In the last years, public and veterinary health authorities and research organizations have heavily invested in the implementation of surveillance plans and the development of systems with the aim of collecting and providing knowledge, data and tools for basic analysis via the web [4,5,6,7,8,9]. Unfortunately, emerging infectious diseases, especially vector-borne diseases, are more challenging to be predicted and controlled, and require a strong multidisciplinary approach due to the interaction among host, vectors, pathogen and environment. Their understanding needs the analysis of complex and diverse data, often disconnected from each other and coming from different sources [10,11]. Especially the environmental component involves the use of data, which although widely available nowadays, are unstructured and extremely large. 

All this makes such data of limited use if not converted through proper data management and analytical methods that can handle the heterogeneous datasets, transforming these into information useful to decision and policy makers [12,13].

However, as new data and computational resources have become available, a wide range of spatial and spatiotemporal methods have been developed for early outbreak detection, cluster detection, identification of risk areas and risk factors and disease transmission pattern evaluation [14,15,16,17]. The application of these increasingly complex statistical methods is fortunately facilitated by the growing development of the open-source community, among which the most widespread and popular is certainly R [18], a programming language and free software environment for statistical computing and graphics. Several R-packages (e.g., *surveillance, sp*, *rSaTScan*, *network*, *tsna*, *igraph*) [19,20,21,22,23,24] have been made available to researchers, but their use still requires adequate programming and statistical skills to write down codes and to perform the analysis effectively. 

To overcome this lack in technical skills, desktop and web applications have been developed, providing analysis tools ready to use for researchers and public and animal health professionals [25,26,27,28,29,30]. Among the others, Nöremark et al., proposed EpiContactTrace, an open source tool, implemented in R-language, to perform contact tracing in real time during disease outbreaks [27]; Moraga et al. developed a Shiny web application, named SpatialEpiApp, that integrates disease mapping and spatiotemporal clusters detection using SaTScan software [26]. In addition, international health organizations are moving in this direction and the European Centre for Disease Prevention and Control (ECDC) for instance, in 2018, launches an interactive Shiny application named EpiSignalDetection, in which the user can import external data and perform basic signal detection analyses.

However, despite the growing effort and the most recent integrated applications, a more extended web platform managing different datasets, and which takes into consideration various analytical tools for a complete epidemiological assessment of disease surveillance data, is still lacking. 

In this paper, we present EpiExploreR, a user-friendly, flexible, R-Shiny web application. EpiExploreR dashboard provides different tools integrating several common approaches used to analyze spatial and spatiotemporal data on animal diseases in Italy, including notified outbreaks, surveillance of vectors, animal movements or contacts data, and remotely sensed data. The dashboard displays and summarizes results at one glance by interactive maps, graphs and tables following the “what you see is what you analyze” scheme. 

This paper is organized as follows: First, we describe the EpiExploreR app aims, its architecture and data flow. Then we briefly introduce the statistical methods and software used in spatial and spatiotemporal analysis. Three case studies are used to illustrate the different applied tools.

## 2. Materials and Methods

### 2.1. EpiExploreR Implementation and Development

EpiExploreR with its interactive, R-Shiny interface allows one to rapidly analyze and visualize data on animal diseases in Italy. It is organized in main topics that can be summarized as follows:Accessing and exploring different sources of geo-referenced, nearly real-time data, including notified outbreaks, surveillance of vectors, animal movements and remotely sensed data;Applying base methods for early outbreak detection (e.g., Farrington algorithm, spatiotemporal cluster analysis and data correlation tools);Running and calibrating temperature-driven mosquito models;Performing network analysis useful in the identification of disease transmission patterns.

Results of data exploration and analysis are displayed through an interactive map, tables and graphs produced in the Map and Analysis section (accessed via the main navigation menu) and the Descriptor section, which creates a customized pivot tables.

To make EpiExploreR available via the Web and allow concurrent usage, ShinyProxy (https://www.shinyproxy.io/) was used. ShinyProxy is the easiest way to deploy Shiny applications, it is open source and makes use of Docker technology. 

ShinyProxy guarantees: (1) a better development-test-production cycle for apps; (2) isolated “workspace” for each session; (3) scalability in the case of requests for more computational resources (through a Docker/container-based cluster). Figure 1 shows the system architecture. All software and R-packages used to develop EpiExploreR application are listed in Table 1.

### 2.2. Data Collection and Data Flow

EpiExploreR integrates complex and diverse public and restricted data, in a single environment, including notified outbreaks, entomological surveillance, animal movements and remotely sensed data coming from different sources.

The epidemiological information is retrieved by the National Information System for the Notification and Management of Animal Diseases in Italy (SIMAN) [58], managed by the Italian Ministry of Health. The details collected for each notified outbreak include the disease, province, administrative unit, outbreak code, latitude and longitude of occurrence, the outbreak state (i.e., suspect, extinct and confirmed), number of cases, susceptible, destroyed, slaughtered and dead animals, date of occurrence and confirmation of the event and species. Entomological surveillance data include the catch site code, latitude and longitude of the catch site, and the number of specimens identified.

Animal movements data are extracted from the Italian National Database for Animal Identification and Registration (NDB) [59], managed by the Italian Ministry of Health. Each movement reports the origin and destination holding code, the number of traded animals per species and the date of the movement. In addition, the following attributes for each holding are considered: the holding type (i.e., farm, slaughterhouse, staging point, market, assembly center, foreign country, etc.), the holding production type (i.e., milk, meat, wool, reproduction, etc.) and the address and geographical coordinates of the holding.

Remotely sensed data are gathered from LP DAAC [7]: MODIS Land Surface Temperature and Emissivity (MOD11A2), Day/Night Land Surface Temperature (LSTD/LSTN) at 1 km spatial resolution and 8 days temporal resolution. 

A daily R-routine gathers and structures data provided from different data sources and sets a default data time-window of the last five years; data are then returned into binary R-data to be efficiently accessible in EpiExploreR.

Users can upload external data (outbreaks, vectors and animal movements or contacts data) using Excel files with a predefined layout downloadable from the application. In addition, EpiExploreR provides a tool for data downloading in Excel file format. Table 2 lists the downloadable data. The system data flow is shown in Figure 1. 

### 2.3. Spatiotemporal Methods and the Epiexplorer Dashboard Features

The EpiExploreR dashboard features are shown in Figure 2, where each task and consequently each tool is linked to the data analysis objective. Although a full description of the provided tools might be useful, for length constraint, only the principal ones are following in detail.

#### 2.3.1. Disease Mapping

An interactive map shows the distribution of outbreaks, vectors or animal movements data (depending on the dataset chosen), in space and time, giving an idea of patterns, disease speed and risk factors. The EpiVelocity tool is provided to estimate the speed of the infectious disease spread using the simpler observed space–time ratio method. Velocity is calculated considering a distance determined by the length of a drawn line on the map, and the time set through a grid with a user-defined resolution, taking into account outbreaks involved in each pixel and their minimum time of occurrence. A time slider is available to dynamically plot the outbreaks on the map.

#### 2.3.2. Early Outbreak Detection (EOD) Methods

The Farrington algorithm is a robust method that can detect the emergence of rare disease outbreaks. It is derived from the outbreak time series over the selected time period, taking into account seasonality and trend of the pattern under study. For each time point (in our case, week) it uses a Poisson generalized linear model (GLM) to predict the number of counts according to the procedure by Farrington et al. [60]. The 95% quantile of this prediction represents the threshold value which the observed distribution is compared to: if the observation is above the boundary, then a warning arises. The upper bound is calculated considering two years back and a three week window. Results are shown for the last 20 weeks in a graph together with the epidemic curve (on a weekly basis) and a smoothed predicted line. Two implementations of the Farrington method are proposed, the classical Farrington [60] and the Farrington Flexible method made by Noufaily et al. [61].

Cluster analysis detection has a long history [62,63] that attracted great interest, and various techniques have been developed so far for evaluating whether the incidence of disease significantly groups together [64,65,66]. The scan statistics methodology [67,68] has been implemented as a major analytical tool for cluster detection in a spatial, temporal and space–time setting. A prospective space–time permutation model is applied to the outbreak cases in the selected time period. This model requires only cases data, with information about the spatial location and time for each case, with no information needed about controls or a background population at risk [64]. The prospective option has been held for the early detection of disease outbreaks, since it regularly scans the current study period for the alive cluster. Users can set the time window (ranging from seven to thirty days) and spatial radius (ranging from five to fifty kilometers) to perform the analysis.

#### 2.3.3. Temperature Driven Mosquito Modeling

The mosquito lifecycle is modeled considering two age compartments: an aquatic stage (eggs, larvae, pupae), and a terrestrial stage (adult mosquitoes). The model, driven by temperature, is performed considering the density-dependent population growth rates of mosquitoes (bounded by the carrying capacity of the mosquito larvae), daily mean temperature (mean value of LSTD and LSTN), and the daytime length at the geographical latitude of the set point (generated by clicking on the map or typing geographic coordinates).

The birth rate for larvae, the transition rate from larvae to adults, the fraction of active mosquitoes and the mortality rate for adults are all described in Rubel et al. [69]. Two distinct U-shape functions for the larvae mortality rate are proposed according to Rubel et al., and Beck-Johnson et al. [70], respectively. The user can evaluate model tuning using custom values for all parameters. The mean daily temperature is derived from applying a smoothing spline procedure onto LST data (after 8 days, missing values have been filled). The documentation about the implemented model is available in the application.

#### 2.3.4. Network Analysis in Livestock Mobility

Animal movement records are usually accepted as a key element for disease prevention and control. The animal movements data are represented as a network, where the holdings of origin and destination are nodes, and the movements of animals are edges. The edges have a direction from origin to destination and a weight defined by the number of animals moved or the number of movements in the timeframe. The topological network structure and the centrality measures of nodes can be used to estimate the disease spread risk and vulnerability based on historical trade patterns. In case of an outbreak, holdings with high centrality can be subjected to targeted in-field investigations and control measures. For example, vaccinating super-spreading nodes may be a more efficient way to control the disease spread than random vaccination [71,72,73,74]. Table 3 describes the static network connectivity properties. An interactive graph of the network is provided to highlight the connected components of the network. The documentation about the implemented network analysis tools is available in the application.

Beside the network properties listed in Table 3, EpiExplorerR allows the user to verify if the scale-free property is satisfied in the network under study. A scale-free network is characterized by a heavy-tailed degree distribution (defined as the probability that any node is connected to *k* other nodes), approximated by a power-law behavior of the form *P(k) ~ k^-γ* with 2< *γ* ≤3. This implies an unexpected, statistical abundance of nodes with very large degrees (i.e., the so-called ‘‘hubs’’ or ‘‘super-spreaders’’). Pastor-Satorass and Vespignani [76] showed that in a scale-free network a large number of individuals can get infected in a finite number of steps. It does not matter that the infected node has a low degree value, it is sufficient that it is just a few links away from a hub.

Temporal path in a dynamic network (Tpath) is a sequence of nodes and edges, such that the onset times of successive elements are greater or equal than those of the previous. The Tpath algorithm performs a time-minimization (earliest arriving path) to find a set of nodes reachable on the forward temporal path from the initial seed [19]. The Forward Reachable Set (FRS) and Backward Reachable Set (BRS) measures, derived from the Tpath analysis, are the set of nodes that can be reachable or reached from an initial seed [77]. These measures identify nodes having the potential, if infected, to infect the greater number of other nodes (risk) or, on the contrary, to be reached from the greater number of nodes (vulnerability) in a dynamic network. The Tpath analysis is proposed to take into account the dynamical nature of animal trade and two applications are provided: The first one to perform a quickly and efficiently trace-back and trace-forward activities from a specified seed and timeframe and the second one to perform the Tpath analysis between two or more selected areas involving more nodes as origins and destinations of movements.

## 3. Results

EpiExploreR is available in two versions: public (https://pub.epiexplorer.izs.it/) and restricted (accessible for internal users). Hereafter, we describe the main features of the application and illustrate their utility through three case studies.

### 3.1. The Estimated Velocity of the BTV-1 Spreading in Central Italy During 2014

Estimating the speed of infectious disease spread is a critical task in epidemiology, and several approaches were adopted in recent years [78,79]. Although sophisticated methods may be used to estimate disease velocity, the simpler observed space–time ratio using the first date of outbreak occurrence can be a useful insight into the formulation of hypotheses in the disease spread investigation. Figure 3 shows the interactive map of the distribution of the bluetongue virus-1 (BTV-1) outbreak in Central Italy in 2014 (color coding time of occurrence). The Outbreaks detection tool visualizes a dynamic report (upper right panel in Figure 3) with the number of outbreaks, the involved species and the subtypes of the virus falling inside the map active viewing. In addition, a graph visualizes the timing of the outbreak (on a weekly basis) and the outbreaks type (farms or other localities).

The EpiCurve task presents the weekly epidemic curve highlighting the BTV-1 2014 epidemic peak in September and a total of 1154 cases (bottom right panel in Figure 3). Moreover, the time slider tool dynamically plots the outbreaks on the map, showing the epidemic onsets in June in Lazio, Sicily, and Calabria regions, spreading further in summer (397 cases), autumn (fall, 675 cases) and winter (82 cases) respectively (Figure 4).

Finally, Figure 5 shows the estimated velocity of the BTV-1 spreading across Central Italy during 2014. By the use of the EpiVelocity task, a grid layer is added on the map. The grid layer has a user-defined spatial resolution of 0.4 decimal degrees, and it is colored on the basis of the minimum occurrence time of outbreaks falling inside each pixel of the grid. Estimated velocity (space/time) along each segment of the drawn line ranges from 0.48 km/day to 2.28 km/day (corresponding to 3.3 km/week and 15.96 km/week), while the overall measured velocity is of 1.23 km/day (about 8.6 km/week), as shown in Figure 5.

### 3.2. West Nile Disease (WND) in Sardinia Region

Cases of West Nile Disease (WND) in Sardinia are reported since 2011, although it is considered endemic since 2014 [80]. In 2018 an increase of WND cases was observed in comparison with the previous four years (Figure 6). The epidemic curve indicates that the number of outbreaks ranged from one to three per week in the disease favorable season until 2017, whilst during 2018 it reached fourteen outbreaks per week. The Farrington algorithm identified the 2018 epidemic as an anomaly (represented by the red line in Figure 6).

A preliminary analysis of temperatures was performed to verify anomalies in 2018, possibly explaining the increased number of cases, and it was observed that the mean of temperatures recorded in May and June 2018 were, on average, about 4 °C lower than the overall average of the five years (Figure 7).

Figure 8 reports the time series of adult mosquitos predicted, along with daily mean temperature for a selected point on the map, using the Rubel. et al. model with default parameter values. The plot shows how that the mosquito model predicted a higher number of adults during 2018. This is a clear evidence of how the 2018 temperature time series was particularly favorable to mosquito’s growth.

Using the vector’s report it is possible to explore correlations between insects species, LST data, outbreaks and simulated mosquito populations for a drawn spatial region on the map. Figure 9 shows the correlation (as scatter-plot and double axes time series) between simulated adult mosquitoes and WND outbreaks falling inside the region drawn on the map on a monthly basis. The results reinforce the hypothesis of favorable temperature conditions for mosquitoes considering the WND outbreak region during the year 2018.

### 3.3. Alive Cluster Detection for Brucellosis Disease and Evaluation of Its Introduction or Spread in Italy through Animal Movements Network Analysis

Searching for alive spatiotemporal disease clusters allows epidemiologists to formulate hypotheses, test associations and find risk factors. By integrating detailed data of livestock displacements and leveraging on the network science approach, it is possible to have a most complete framework to study real epidemic outbreaks to identify where and when infection could move [81,82].

Using the SatScan tool for the identification of Brucellosis alive clusters in Italy in April 2019, two significant clusters were detected in northern Sardinia (*p*-value < 0.05) with 29 outbreaks between March and April 2019 (Figure 10).

Network analysis tools applied to bovine and sheep movement data in the period 01/01/19–30/03/19 preceding the time of clusters detection and related to the identified area, shows that the majority of outgoing movements were registered towards northern Italy, which is disease-free (Figure 11). Moreover, the map and graph of Figure 11 show the absence of incoming direct connections between the selected areas and the Campania and Calabria regions, Apulia and Sicily, where brucellosis is endemic [81,82]. This probably links the cluster origin observed in Sardinia to a trade activity inside the region, rather than a legal introduction from high prevalence Italian regions.

The topological properties of the network show that it has strong diffusive capacities in terms of epidemic size and speed (Figure 12). The degree distribution of nodes, following a power law distribution with exponent 2.95, implies the presence of super-spreaders nodes. In addition, the average shortest path length value indicates the possibility for the disease of reaching any other farms in the network from any seeding site with a sequence of around two steps [76].

Figure 13 shows a table listing centrality measures for each node of the selected network. The table is linked to the map and the network graph so that the nodes with the highest degree centrality value can be easily spatially and graphically identified. These nodes can be subjected to targeted in-field investigations and control measures [70,71,72].

Tpath analysis of the selected network data can be used to take into account the dynamical nature of animal trade, and consequently the disease spread. Indeed, the time window of a possible introduction of infection to the farms is relevant when determining contacts of interest. Animals introduced after the possible window of introduction can be excluded as a source, and animals leaving the farms before the possible introduction will not have the chance to spread the disease. Figure 14, panel A, shows an example of trace forward starting from a seed node, within the identified cluster region, and reaching northern Italy in a time window of possible infection introduction (from 01/12/2018 to 28/02/2019), while the geographical distribution of farms reached by the outgoing animal movements through temporally valid paths from the selected areas is shown on the map in panel B.

The Tpath tool can be used to evaluate all temporal valid paths (either direct or indirect) from an origin area to a destination area (spatially identified). Figure 15 shows the Tpath analysis setting a 10 km circular buffer around Piacenza (an example of the disease-free zone in northern Italy) as a destination area. Results show that this Piacenza area has been reached by several temporal valid pathways. However, these connections involved at least five intermediate steps and an elapsed time of 20 days. Intermediate points (so-called bridges) connecting Sardinia to Piacenza are all located in northern Italy, mostly on the western side.

## 4. Discussion

For national health administrations, the availability of epidemiological georeferenced data, along with potential disease or vector risk factors, contacts trace, genomic sequences, has drastically increased in recent years. Public health administrations and research institutes are constantly focused on the adoption of open-source software for the management, analysis and distribution of these data. R software is certainly the most used, it is a free environment for statistical computing, flexible and continuously supported by a large community of active developers and researchers. In this context, “R-epi project” (https://sites.google.com/site/therepiproject/), is a pillar example of a website developed to provide resources for the analysis of disease data using R.

In addition, the recent development of the shiny package made easier the development of web applications from within R, and consequently a number of shiny apps for the epidemiology of infectious diseases have been lately developed [25,26,27,28,29,30]. However, a more extended web platform managing different datasets, and which takes into consideration proper analytical tools for a complete epidemiological assessment of disease surveillance data, is still lacking.

EpiExploreR has been implemented using R and Shiny, and includes functions from different R packages and statistical programs. The application is user friendly and is addressed to scientists, researchers, including policy makers and public and animal health professionals wishing to test hypotheses and explore data on disease surveillance activities, even if not statistically or programming skilled.

The system provides data and analysis features, and is mainly (but not only) designed to efficiently manage, visualize and analyze the huge amount of diverse epidemiological data.

It is important to note that the loading data tool allows choosing among:Load example data to explore the complete set of features for training purposes;Load external data to perform the analysis using self-owned data (in both the public and private versions);Load data provided by national data sources: NDB and SIMAN (only for private version).

The loading of MODIS LST data provided by LP-DAAC is carried out in each case and limited to Italy.

What makes EpiExploreR advanced and original is the adoption of a “what you see is what you analyze” scheme. The interaction between spatiotemporal data filters, analyses and exploration tools allows the analysis to be refined, step by step, easily and iteratively, and respond to the needs on different temporal and geographical scales or user-defined areas. Furthermore, the users can run SatScan software to detect spatiotemporal clusters. Although applications with similar features have been developed (e.g., SpatialEpiApp), these are mostly focused on one of the many factors related to diseases, whilst EpiExplorer integrates several factors.

In some cases, the analysis can lead to cumbersome results (if a huge amount of data freezes the browser) or to long waiting time. Aware of the idea that these aspects can be problematic for web applications, we left, however, to the user the choice (e.g., the plot of a large network, is limited to 1000 edges by default, but the user can force its rendering).

Actually, the implemented methods are commonly used in health surveillance, but the application can be easily extended in future versions. Several aspects will be improved, mainly to:Upload an additional dataset (e.g., genomic sequences and animal density data) and use the appropriated spatiotemporal and mathematical models.Create custom epidemiological reports in HTML-format. Include tools to import and export spatial data (e.g., shapefiles) reproducibility of the performed analyses.

These upgrades will ensure an increase in the flexibility of the system and greater reproducibility of the performed analyzes.

## Figures and Tables

**Figure 1 microorganisms-07-00680-f001:**
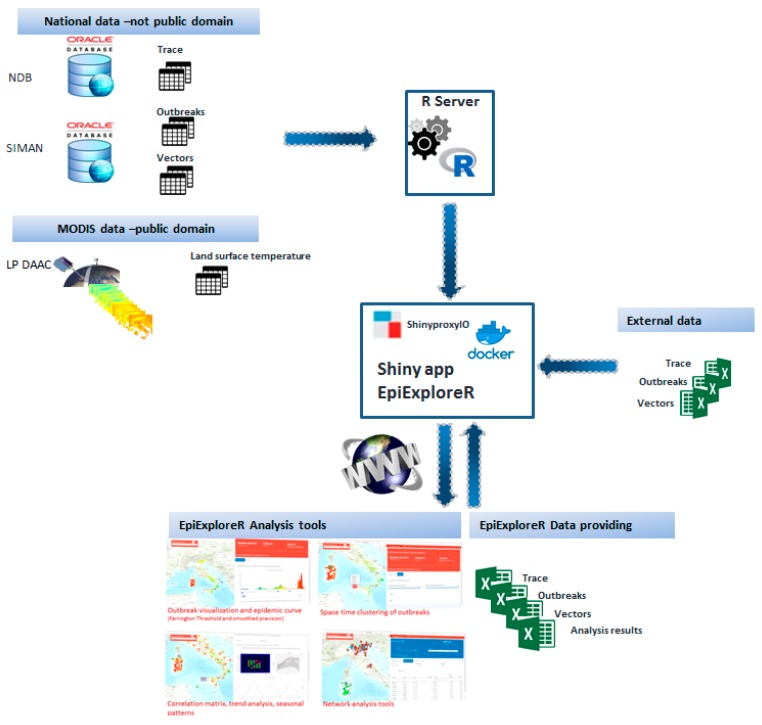
EpiExploreR architecture and data flow.

**Figure 2 microorganisms-07-00680-f002:**
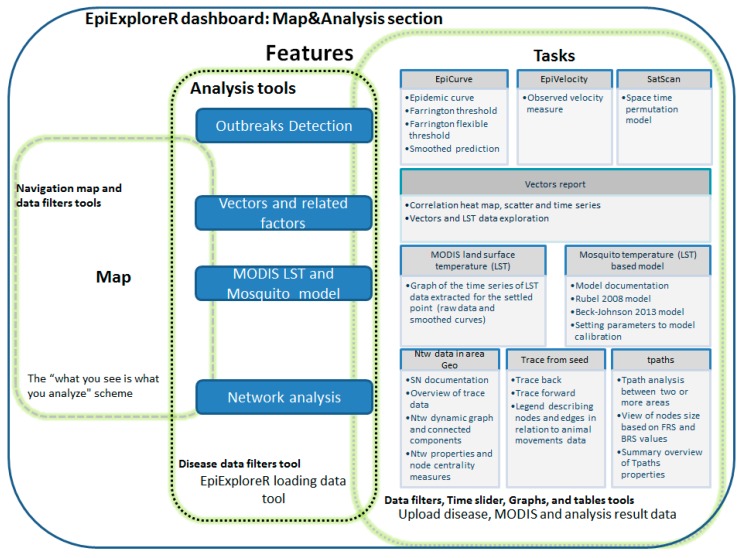
EpiExploreR dashboard features scheme.

**Figure 3 microorganisms-07-00680-f003:**
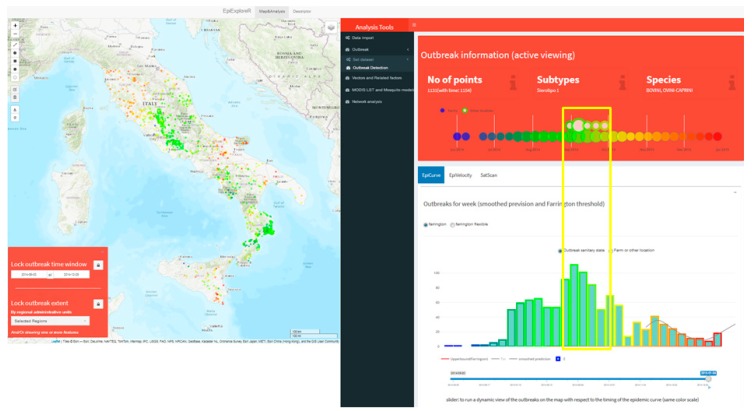
Bluetongue virus-1 (BTV-1) outbreaks distribution in Central Italy in 2014. The weekly epidemic curve highlighting the BTV-1 2014 epidemic peak in September (yellow rectangle in the dashboard).

**Figure 4 microorganisms-07-00680-f004:**
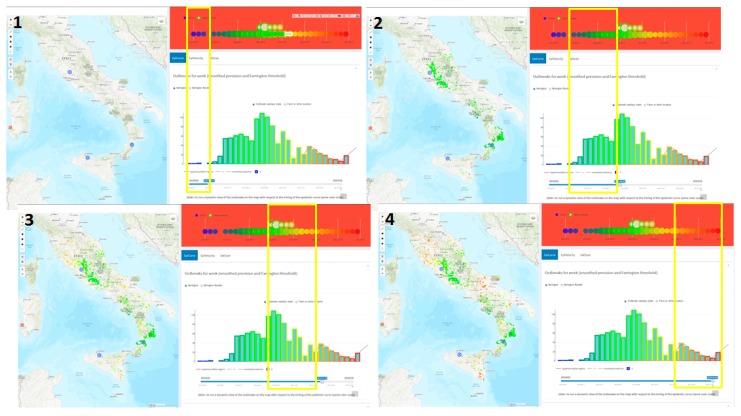
BTV1 epidemic spread in Italy in 2014 as showed using the time slider tool in four periods: onset in June (1), Summer (2), Autumn (3) and Winter (4). The yellow rectangle defines the time period.

**Figure 5 microorganisms-07-00680-f005:**
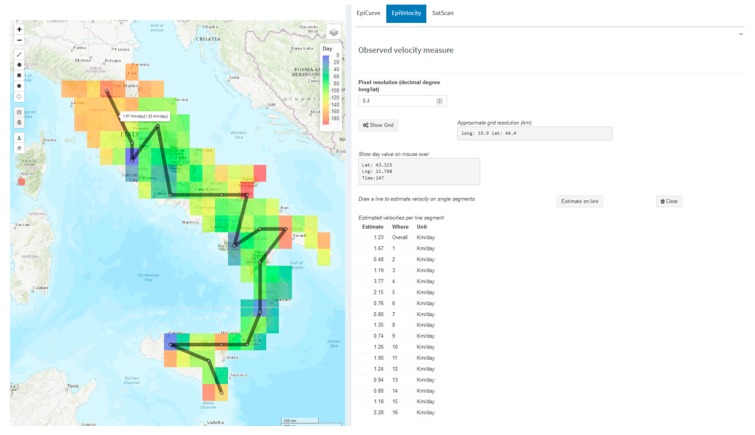
The grid layer represents the minimum occurrence time of outbreaks falling inside each pixel of the grid. Estimated velocity of the BTV-1 spreading in Central Italy in 2014. The overall and single segment velocities are reported in the bottom right table.

**Figure 6 microorganisms-07-00680-f006:**
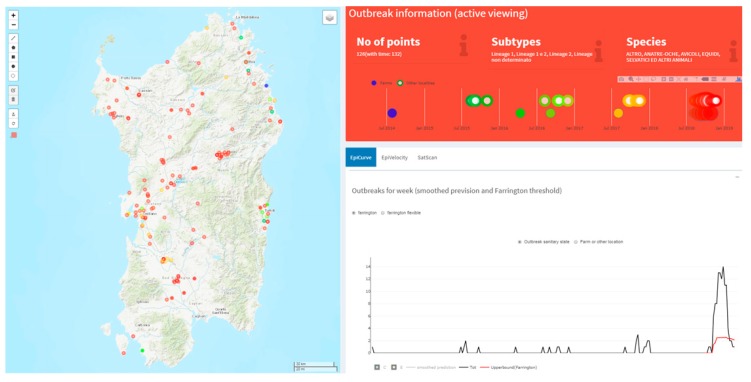
Epidemic curve of 2014–2018 West Nile Disease (WND) outbreaks and Farrington threshold. Farrington algorithm identifies the 2018 epidemic as an anomaly, for the whole favorable season.

**Figure 7 microorganisms-07-00680-f007:**
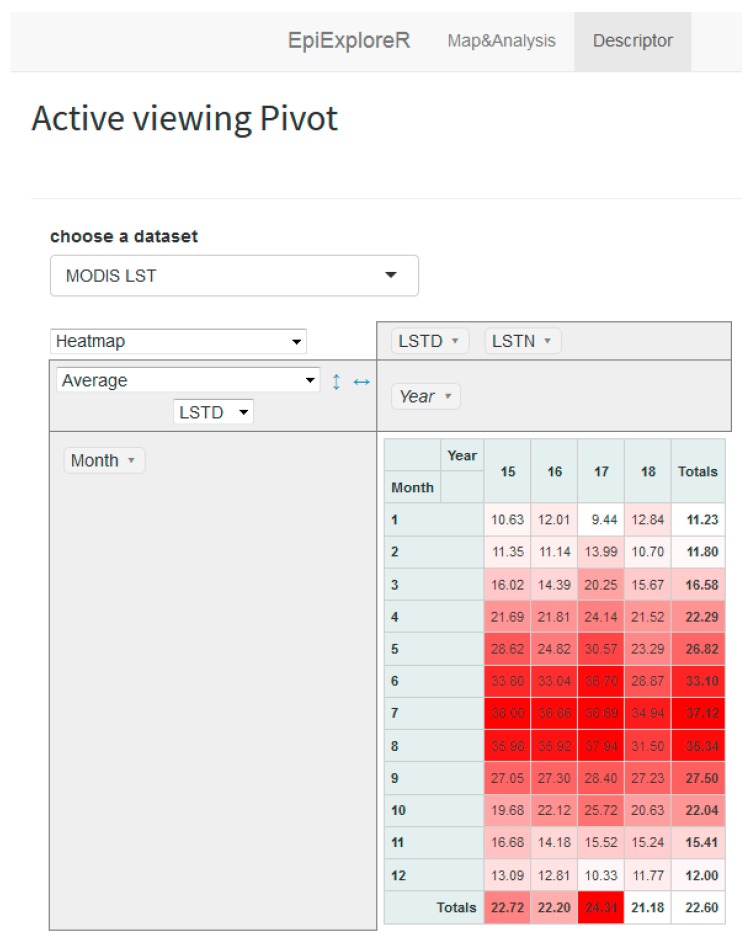
The Pivot table, in the “descriptor” tab, represents the average of the day-time land surface temperature per month and year in the last five years (data refers to the active map view). The table shows that May (23.29 °C) and June (28.87 °C) temperatures were about four degrees lower than the overall five years averages (26.82 °C and 33.10 °C respectively).

**Figure 8 microorganisms-07-00680-f008:**
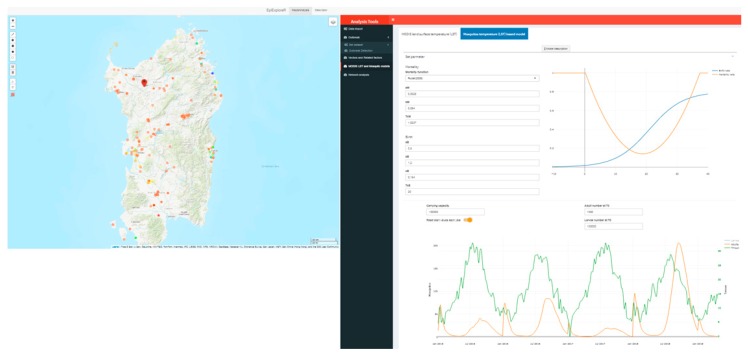
Time series of adult mosquitos (orange line), along with daily mean temperature (green) for the selected point (red marker on the map) using the Rubel. et al. model. The map is that of the Italian island of Sardinia.

**Figure 9 microorganisms-07-00680-f009:**
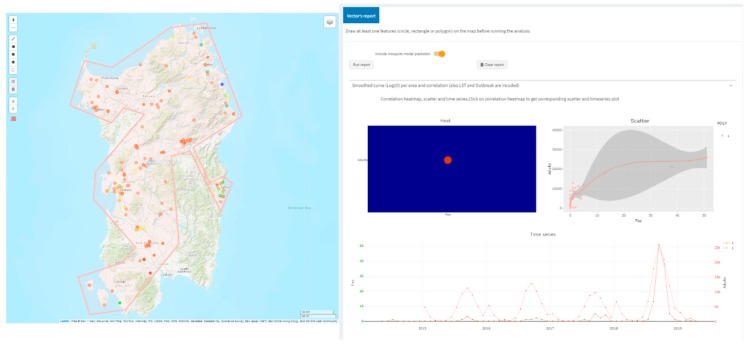
Correlation (as scatter-plot and double axes time series) between simulated adult mosquitoes and WND outbreaks falling inside the (pink) region drawn on the map on a monthly basis for 2014–2018 years. Again, the map is that of the Italian island of Sardinia.

**Figure 10 microorganisms-07-00680-f010:**
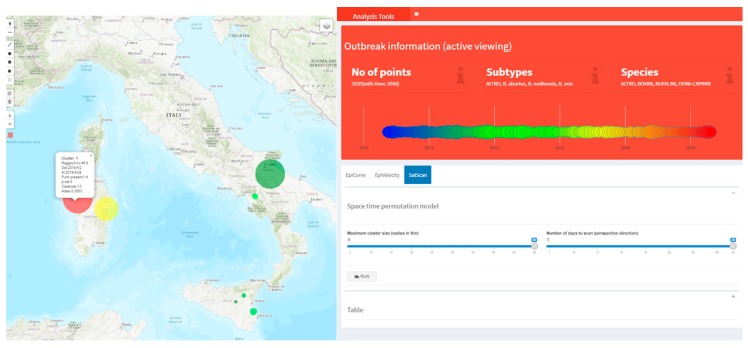
The SatScan tool results (as seven colored clusters on the map) obtained by setting 30 days for the time window and a radius of 50 km in the parameter setting tab (on the right). The outbreak layer has been hidden to make the smallest clusters visible. Cluster shape color is *p*-values coded from red to yellow when *p* is less than 0.05, and from light to dark green when *p* is greater than or equal to 0.05. Clicking on a cluster on the map opens a popup, showing further details such as the time window, the radius, *p*-value, the observed and expected number of outbreaks.

**Figure 11 microorganisms-07-00680-f011:**
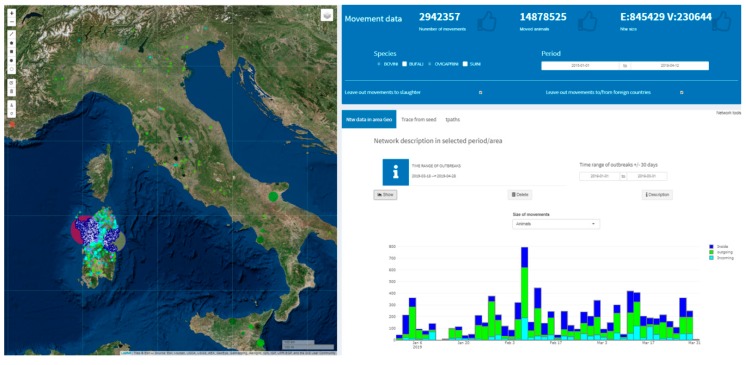
The distribution of farms (nodes) involved in the network are mapped (left side of the figure) using colors to differentiate nodes outside the selected area in three categories: reaching (light blue dots), reached by (green dots) and both reaching and reached by (pink dots) the selected area (clusters). Nodes inside the selected area are represented by dark blue points (blue and white markers highlight nodes involved in trading, only inside the area). The report produced by ‘Ntw data in area geo’ task includes a bar chart of the animals traded through time with the distinction of outgoing (green), incoming (light blue) and inside (dark blue) the areas drawn on the map. Jointly looking at the map and the graph remarks that most of the animals going to selected areas come from the Sardinia region (light blue points on the map).

**Figure 12 microorganisms-07-00680-f012:**
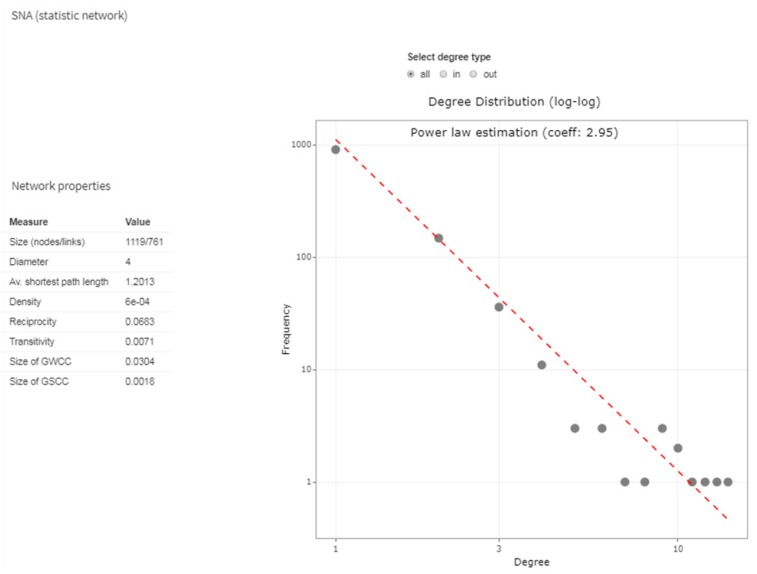
Topological properties of the static network are shown in table and graph form. In detail, the table reports the network size, diameter, average shortest path length, density, reciprocity, transitivity and the Giant Weak and Strong Connection Component percentage size, respectively. The graph reports the Degree distribution of nodes.

**Figure 13 microorganisms-07-00680-f013:**
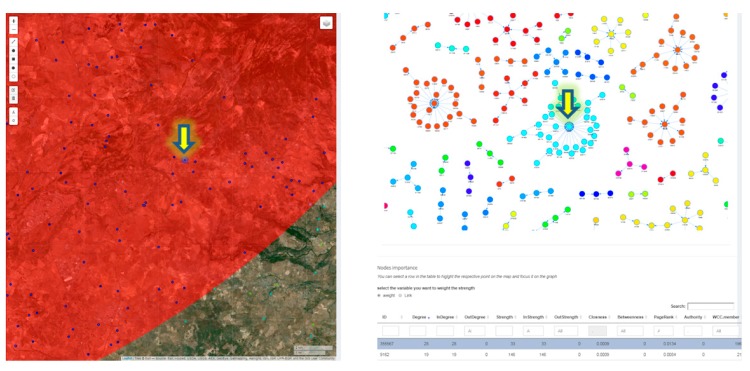
Network local properties for each node (degree, indegree, outdegree, strength, instrength, outstrength, closeness, betweenness, authority, PageRank) are listed in the table. The selected record in the table is linked to the corresponding node (the yellow arrow) on the map and to the dynamic graph of the network (color coding weakly connected components).

**Figure 14 microorganisms-07-00680-f014:**
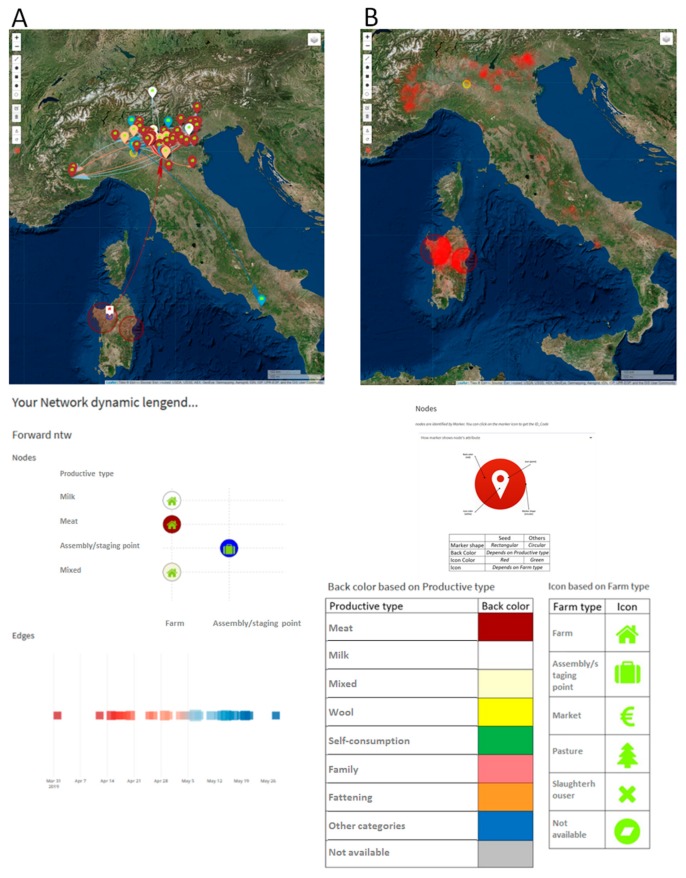
Panel A shows an example of the trace forward subnetwork linking a node in Sardinia to northern Italy in the time window 01/12/2018–28/02/2019. The forward network representation adds details about the timing of the edges, the number of animals moved and the type of farms involved (coded through the marker’s color and icon, as detailed in the legend). Panel B shows country-wise reached nodes through valid temporal paths from the Sardinia areas (red points transparency can be set to get a density-like map).

**Figure 15 microorganisms-07-00680-f015:**
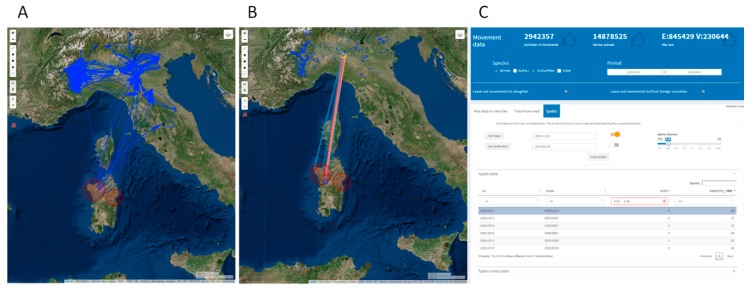
Panel A shows the subnetwork originating in nodes belonging to the red circle area and reaching nodes inside the yellow circle area through temporally valid paths. In panel B the subnetwork found is summarized using direct straight-line connecting nodes, whose color reflects the arrival time. Node size inside the origin area reflects the number of reached nodes inside the destination area (similarly, node size inside the destination area reflects the number of nodes they are reached by). The map also shows nodes (or bridges) crossed by the temporal paths (blue points), whose size represents the degree value of the node. Panel C displays network characteristics, parameter settings and the Tpaths tables. Tables are linked to the map and they list path properties, like origin and destination farm code, the number of steps and the time of earliest arriving.

**Table 1 microorganisms-07-00680-t001:** Software and R packages used to develop EpiExploreR.

	Ref.	Description	Tool or Task
**Software**			
R	[18]	Language and environment for statistical computing and graphics.	User interface
SaTScan	[31]	Software that analyzes spatial, temporal and space-time data using scan statistics.	SatScan
**R-package**			
surveillance	[20]	Temporal and Spatiotemporal Modeling and Monitoring of Epidemic Phenomena.	EpiCurve
shiny	[32]	Web Application Framework for R.	User interface
shinyjs	[33]	Perform common useful JavaScript operations in Shiny apps that will greatly improve the apps without having to know any JavaScript.	User interface
shinydashboard	[34]	Create dashboard with Shiny.	User interface
shinythemes	[35]	Themes for Shiny.	User interface
shinyWidgets	[36]	Custom Inputs Widgets for Shiny.	User interface
shinycssloaders	[37]	Add CSS Loading Animations to ‘Shiny’ Outputs.	User interface
sp	[21]	Classes and Methods for Spatial Data.	Multiple
rsatscan	[22]	Tools, Classes and Methods for Interfacing with SaTScan Stand-Alone Software.	SatScan
network	[23]	Tools to create and modify network objects.	Ntw data in area Geo
tsna	[19]	Temporal SNA tools for continuous- and discrete-time longitudinal networks.	Trace from seed and TPath
visNetwork	[38]	It allows an interactive visualization of networks.	Ntw data in area Geo
igraph	[24]	Routines for simple graphs and network analysis.	Ntw data in area Geo
leaflet	[39]	Create and customize interactive maps using the ‘Leaflet’ JavaScript library and the ‘htmlwidgets’ package.	User interface
raster	[40]	Reading, writing, manipulating, analyzing and modeling of gridded spatial data.	EpiVelocity
plotly	[41]	Create Interactive Web Graphics via ‘plotly.js’.	Graphs
ggplot2	[42]	Create Elegant Data Visualizations Using the Grammar of Graphics.	Graphs
rpivotTable	[43]	Build Powerful Pivot Tables and Dynamically Slice and Dice your Data.	Descriptor section
dplyr	[44]	A fast, consistent tool for working with data frame-like objects, both in memory and out of memory.	Multiple
emojifont	[45]	An implementation of using emoji and fontawesome for using in both base and ‘ggplot2’ graphics.	Ntw data in area Geo
RColorBrewer	[46]	Provides color schemes for maps.	Map
DT	[47]	Data objects in R can be rendered as HTML tables using the JavaScript library ‘DataTables’ (typically via R Markdown or Shiny).	Tables
rgdal	[48]	Bindings for the ‘Geospatial’ Data Abstraction Library.	Multiple
bezier	[49]	Toolkit for Bezier Curves and Splines.	Ntw data in area Geo
leaflet.extras	[50]	Extra Functionality for ‘leaflet’ Package.	User interface
rgeos	[51]	Interface to Geometry Engine—Open Source (‘GEOS’).	Multiple
mgcv	[52]	Mixed GAM Computation Vehicle with Automatic Smoothness Estimation.	EpiCurve
v8	[53]	An R interface to Google’s open source JavaScript engine.	Multiple
xlsx	[54]	Read, Write, Format Excel 2007 and Excel 97/2000/XP/2003 Files.	Data download/upload
RCurl	[55]	General Network (HTTP/FTP/...) Client Interface for R.	Data download/upload
htmlwidgets	[56]	A framework for creating HTML widgets.	User interface
stats4	[18]	Statistical Functions using S4 classes.	Multiple
ggmap	[57]	A collection of functions to visualize spatial data and models on top of static maps from various online sources (e.g., Google Maps and Stamen Maps).	Map

**Table 2 microorganisms-07-00680-t002:** EpiExploreR downloadable data.

App Analysis Tool/Task	File Name	Worksheet Name	Data Description *
Outbreak detection/EpiCurve	outbreak.data.xlsx	Outbreaks	Outbreak disease data
Vectors and related factors /Vectors report	ento.data.xlsx	Ento	Entomological data
Vectors and related factors /Vectors report	ento.data.xlsx	Outbreaks	Outbreak disease data
Vectors and related factors /Vectors report	ento.data.xlsx	LST_RAW	LST data (at 8 day temporal resolution)
Vectors and related factors /Vectors report	ento.data.xlsx	LST_Month	LST data (monthly temperature average)
MODIS LST and Mosquito model/MODIS Land surface temperature (LST)	LSTpoint.data.xlsx	PointCoordinates	Coordinates of the user-defined point
MODIS LST and Mosquito model/MODIS Land surface temperature (LST)	LSTpoint.data.xlsx	LST.Observed	TLS data for the set point (8 days temporal resolution values)
MODIS LST and Mosquito model/MODIS Land surface temperature (LST)	LSTpoint.data.xlsx	LST.interpolation	Interpolated daily TLS values
MODIS LST and Mosquito model/MODIS Land surface temperature (LST)	LSTpoint.data.xlsx	LST.NA	TLS data missing
MODIS LST and Mosquito model/MODIS Land surface temperature (LST) based model	MosquitoModel.xlsx	MosquitoModel	Mosquito model results: Larvae/Adults daily data and related mean temperature values
Network Analysis/Ntw data in area Geo	NodeCentralities.xlsx	Nodes Centralities	Nodes data and related centrality measures values
Network Analysis/Ntw data in area Geo	NodeCentralities.xlsx	Static edges	Contacts data of the static network
Network Analysis/Ntw data in area Geo	NodeCentralities.xlsx	Nodes	Nodes data of the static network
Network Analysis/Trace from seed	Subntw.xlsx	TraceFromSeed	Movements data related to the specified seed in back and forward in the established timeframe
Network Analysis/Tpaths	tpaths.Tables.xlsx	Selection	Data related to the Tpath analysis: start/end date, species, (from/to) slaughter/foreign state movements (included/excluded)
Network Analysis/Tpaths	tpaths.Tables.xlsx	Tpath edges table	Tpath analysis results in terms of edges involved
Network Analysis/tpaths	tpaths.Tables.xlsx	Tpath nodes table	Classification of nodes included in the Tpath analysis in terms of their FRS values (origin area), BRS values (destination area) and DEG values for bridge nodes (external to the origin and destination areas)

***** Downloaded data are referred only to the space-temporally data defined by the user.

**Table 3 microorganisms-07-00680-t003:** Network properties implemented in EpiExploreR.

Name	Description *
**Network Properties at the Global Level**	
Size	The number of nodes and edges.
Diameter	The length of the longest path (in number of edges) between two nodes.
Average shortest path length	Refers to the average of all the shortest distance (number of edges) between each pair of reachable nodes in the network [75].
Density	The number of edges in the network over all the possible edges that could exist in the network.
Reciprocity	Measures the mutual edge relation: the probability that if node *i* is connected to node *j,* node *j* is also connected to node *i*.
Transitivity	Measures that probability that adjacent nodes of a network are connected (also known as clustering coefficient).
Network communities	The networks often have different clusters or communities of nodes that are more densely connected to each other than to the rest of the network.
**Network Properties at Local Level (the Weighted Measures are Calculated Considering as Edge Weight Alternatively the Number of Animals Moved or Number of Movements)**	
Degree	The number of adjacent edges to each node. It is considered as InDegree and OutDegree: InDegree is a count of the number of incoming edges to the node and OutDegree is the number of outgoing edges from the node.
Strength	A weighted measure of degree that takes into account the number of edges going from one node to another or the number of animals moved.
Closeness	Measures how many steps are required to access every other node from a given node.
Betweenness	The number of shortest paths between nodes, passing through a particular node.
Page rank	Approximates the probability that any message will arrive to a particular node.
Authority score	A node has high authority when it is linked to many other nodes, in turn linked to many other nodes.

* http://pablobarbera.com/big-data-upf/html/02b-networks-descriptive-analysis.html.

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
