# Peer review of "EpiExploreR: A Shiny Web Application for the Analysis of Animal Disease Data"

_microorganisms, 2019, doi:10.3390/microorganisms7120680_

Round 1

Reviewer 1 Report

The authors developed a web application for the analysis of animal disease data and presented all the technical details about the web application they have designed. The manuscript, for the most part, discusses data curation, design, integration, and usage of their application for knowledge inference in the related domains. Such risk detection and surveillance platform deemed useful for public health authorities in policy decision making and also for the wider scientific audience.

This is generally a well-written manuscript and the authors have presented sufficient details related to the design and usage of their application. The application offers an interesting data analytic platform for modern era data-driven animal disease surveillance. A few technical and non-technical issues, however, need to be addressed that I believe would aid authors in improving the quality of their presented work.

A few aspects of the application are little confusing from the users’ perspective but I hope that all such changes and cosmetic improvements could occur in future versions as the developed application will mature over time. Also, as pointed by the authors too, the web application is much slower for large datasets, this may be partially due to the choice of hosting platform and the available computational resources, therefore a warning note on the application for such cases might be helpful to make users aware of this fact. Page 9 and line 194, the link provided by the authors is not functional (it says “File not found”). Please make sure all the links provided in the manuscript are accurate and fully functional. The text is figure 14 is not readable. If possible, please enhance the text size within the figure to make it understandable.   

Author Response

A few aspects of the application are little confusing from the users’ perspective but I hope that all such changes and cosmetic improvements could occur in future versions as the developed application will mature over time.

We thank the reviewer for the useful suggestions. Surely, it will be our next task to collect users' feedback to make appropriate changes and improvements to the application and make it more user-friendly in its coming versions.

Also, as pointed by the authors too, the web application is much slower for large datasets, this may be partially due to the choice of hosting platform and the available computational resources, therefore a warning note on the application for such cases might be helpful to make users aware of this fact.

This is particularly true for the network analysis tools. As suggested by the reviewer we have added a warning message in the ‘Network Analysis’ section: “Please pay attention to the size of the selected dataset. Large dataset might take longer for being analysed ".

Page 9 and line 194, the link provided by the authors is not functional (it says “File not found”). Please make sure all the links provided in the manuscript are accurate and fully functional.  

We accessed again the current link: http://pablobarbera.com/big-data-upf/html/02b-networks-descriptive-analysis.html and it works in our browsers 

The text is figure 14 is not readable. If possible, please enhance the text size within the figure to make it understandable.   

We have modified Figure 14 to make it more readable.

Reviewer 2 Report

The Shiny tool described in this manuscript implements many useful analyses and integrates with geographic information. I feel like the science is fine, although future technical improvements are needed for a larger impact.

It will be more interesting to broad users if it supports regions other than Italy. The project may focus on private collaboration initially, but most functionalities are applicable with user provided geo coordinates as well.

The web interface only works well on screens with very large resolutions like retina. Because it tries to fuse the map with dashboard side by side with fixed positioning and width percentages, without considering responsive design (e.g. fluidRow in Shiny). On an ordinary FHD screen, many elements overflow and table columns are out of screen, etc. Not to mention any mobile devices. The toolbar tooltip on the map is too transparent to see the label.

In Table 3, the descriptions should be capitalized.

Author Response

It will be more interesting to broad users if it supports regions other than Italy. The project may focus on private collaboration initially, but most functionalities are applicable with user provided geo coordinates as well.

We thank the reviewer for the useful suggestions and we will extend the application to other regions in the future as far as satellite data is concerned. EpiExploreR currently can host non-Italian data (using the templates available in the application) and many analyses can be successfully run (i.e. network analysis tools), but the satellite image size for a global application is definitely an issue. 

The web interface only works well on screens with very large resolutions like retina. Because it tries to fuse the map with dashboard side by side with fixed positioning and width percentages, without considering responsive design (e.g. fluidRow in Shiny). On an ordinary FHD screen, many elements overflow and table columns are out of screen, etc. Not to mention any mobile devices. The toolbar tooltip on the map is too transparent to see the label.

The reviewer's consideration is correct and in the next realise, we will be more careful in adapting the interface to the different monitor resolutions using objects that do not have a fixed position. However, the application was conceived as a dashboard for analysis and research and requires technology more suited to its use by the user.

The toolbar tooltip on the map is too transparent to see the label.

The toolbar tooltip on the map has been modified

In Table 3, the descriptions should be capitalized.

Done